# There Is No Evidence That Inactivated COVID-19 Vaccines Increase Risks of Uveitis Flare

**DOI:** 10.3390/vaccines10101680

**Published:** 2022-10-08

**Authors:** Hang Song, Chan Zhao, Meifen Zhang

**Affiliations:** Peking Union Medical College Hospital, Beijing 100730, China

**Keywords:** COVID-19, inactivated vaccine, uveitis flare

## Abstract

This is a retrospective study to investigate the impact of inactivated Coronavirus disease-2019 (COVID-19) vaccination on uveitis flare in patients with uveitis. Sixty patients that were regularly followed up for uveitis for at least two months after the last dose of inactivated COVID-19 vaccines were included in the vaccination group. Sixty patients with comparable characteristics of uveitis who had not received the COVID-19 vaccines were included in the control group. Uveitis flare within 30 days and 60 days after the vaccination in the vaccination group, or after a randomly selected date in the control group, were statistically compared. The flare rate was 16.7% (30 days) and 23.3% (60 days) in the vaccination group, while it was 13.3% (30 days) and 25% (15/60) in the control group. There was no statistical difference in the flare rate of uveitis between the two groups (*p* = 0.471 for 30 days, *p* = 0.347 for 60 days). Inactivated COVID-19 vaccination appeared not to increase the flare rate in patients with uveitis. Ophthalmologists should give proper and individualized recommendations based on the overall conditions of patients.

## 1. Introduction

Coronavirus disease-2019, abbreviated as COVID-19, has been a global pandemic since 2020. Up to October 2022, 68% of the world population has received at least one dose of the COVID-19 vaccine [1], and there have been numerous reports on the safety and efficacy of different types of COVID-19 vaccines [2,3,4,5,6]. However, knowledge on the safety of inactivated COVID-19 vaccines on patients with uveitis is limited, despite this population being of special concern. On the one hand, immunosuppressive treatments may increase their vulnerability infections [7], possibly including COVID-19; On the other hand, it is concerned that COVID-19 vaccines may provoke uveitis flare in patients with quiescent uveitis because they were reported to be associated with new onsets of uveitis [8,9,10,11]. Theoretically, vaccines might trigger autoimmunity by molecular mimicry between vaccine peptide fragments and uveal self-peptides, immune complex deposition in delayed-type hypersensitivity, and immune reaction in response to vaccine adjuvants [12,13]. However, epidemiological studies do not support the hypothesis that vaccines cause systemic autoimmune diseases [14,15,16], except for the rare associations between the flu vaccine and Guillain-Barré syndrome [17], measles–mumps–rubella vaccine (MMR), and thrombocytopenia [18,19]. Furthermore, adverse reactions caused by vaccines, if any, are much lower in frequency and severity than those caused by spontaneous infection [20,21] if not vaccinated. Thus, clinicians are responsible for giving deliberate and authoritative instructions about vaccinations to these patients. Although there were some cases of new onset of uveitis after inactivated COVID-19 vaccines reported [10,22,23,24,25,26], the causality should be determined by more convincing well-designed studies. The current study aimed to evaluate the risk of uveitis reactivation upon inactivated COVID-19 vaccination in the Chinese population with non-infectious chronic or recurrent uveitis.

## 2. Materials and Methods

COVID-19 vaccination records were reviewed and documented in patients with non-infectious chronic or recurrent uveitis who visited the uveitis outpatient clinic at Peking Union Medical College Hospital between 9 August 2021 and 31 May 2022. Patients who had been regularly followed up for at least one year before vaccination and two months after the last dose of COVID-19 vaccine were included in the vaccination group, and those who had been regularly followed up for longer than 2 years but had not been vaccinated were included in the control group. The date of the last COVID-19 vaccination was designated as the “vaccination date” of the vaccination group, while a random date around 8 months before their last follow-up visit was defined as the presumed “vaccination date” for the control group. Uveitis flare was defined as the emergence of any of the following manifestations: fresh keratic precipitates; ≥0.5+ anterior chamber cells score; ≥0.5+ vitreous haze score; the presence of active fundus manifestations (e.g., macular edema, diffuse choroiditis, bullous serous retinal detachments, focal areas of subretinal fluid, retinal hemorrhages, retinal infiltrates). Demographic and clinical data collected included patients’ age, sex, course of disease, etiologies, anatomical classifications, best-corrected visual acuity (BCVA) of the affected eye at the quiescent stage, and systemic treatment regimens for uveitis.

The terminology and classification of uveitis were in accordance with the Standardization of Uveitis Nomenclature working group [27]. BCVA was converted to the equivalent logarithm of the minimum angle of resolution acuity (logMAR) [28], with no light perception (NLP) was set at 2.9 logMAR, light perception (LP) at 2.6 logMAR, hand movements (HM) at 2.3 logMAR, and counting fingers (CF) at 1.85 logMAR [29]. Systemic treatment regimens included oral corticosteroids and immuno-suppressive treatments such as methotrexate (MTX), mycophenolate mofetil (MMF), azathioprine (AZA), and cyclosporine (CsA), Tacrolimus (TAC), biologic therapies such as interferon-α (INF-α), or adalimumab. Uveitis flare within 30 days and 60 days after vaccination were documented dichotomously and compared between each group. Uveitis flare within 60 days after vaccination in the vaccination group was critically analyzed.

Statistical analysis was performed using Statistical Package for the Social Sciences (SPSS, Chicago, IL, USA) version 23.0. A quantile–quantile plot was used for a test of normality. Age, BCVA are presented as the mean ± standard deviation (SD) and analyzed with Student’s *t*-test. Other demographic and clinical parameters are demonstrated by proportions and analyzed with chi-squared test or Fisher’s exact test if not appropriate. *p*-value of less than 0.05 was considered significant.

The study adhered to the tenets of the Declaration of Helsinki and was approved by the Institutional Review Board of Peking Union Medical College Hospital (K22C0248).

## 3. Results

### 3.1. Demographic and the Clinical Features of Uveitis Patients

Sixty patients were included in the vaccination group and 60 patients were included in the control group. The average age was 38 ± 14 years old and the male to female ratio was 1 (60:60). The mean disease course before vaccination or presumed vaccination date was 5.43 ± 6.05 years. The demographic feature and the clinical feature of uveitis in each group are demonstrated in Table 1.

### 3.2. Systemic Treatment Regimens in Each Group

In the vaccination group, at the time of vaccination, 39 patients had no systemic treatment; 3 patients had systemic steroids only; 14 patients were under immunosuppressant therapy, of which 9 patients had combined with systemic steroids treatment and 1 patient had combined with systemic steroids and INF-α treatment; 4 patients were treated with adalimumab alone. In the control group, at the time of the presumed vaccination date, 19 patients had no systemic treatment; 4 patients had systemic steroids only; 27 patients were under immunosuppressant therapy, of which 23 patients had combined with systemic steroids treatments; 6 patients were treated with adalimumab, and 4 patients were under treatment of INF-α with or without other immunosuppressives.

### 3.3. Vaccination Overview

The type of COVID-19 vaccine administered in the vaccination group were all inactivated vaccines, including BBIBP-CorV (Sinopharm/BIBP, Beijing, China), CoronaVac (Sinovac Biotech, Beijing, China), and Zifivax (Anhui Zhifei Longcom, Anhui, China). Of the 60 patients who had been vaccinated at least once, 52 patients received a second dose, and 13 patients received a third dose. All the sequential doses were from the same manufacturer as the first dose vaccine the patient had received.

### 3.4. Flare Rate and Individualized Analysis

The 30-day flare rate was 16.7% (10/60) in the vaccination group and 13.3% (8/60) in the control group. The 60-day flare rate was 23.3% (14/60) in the vaccination group and 25% (15/60) in the control group. Statistical analysis showed no significant difference in the flare rate between each group in the 30-day follow-up (*p* = 0.609) or the 60-day follow-up (*p* = 0.831). Within the vaccination group, 42.8% cases of uveitis flare were from females. There is no statistically significant difference in the flare rate between patients who were under systemic immunosuppressive therapy or patients who were not (*p* = 0.471 for 30 days, *p* = 0.347 for 60 days). For the 14 patients who had the flare, 9 patients had the flare after the second and 3 patients after the third dose.

Fourteen patients had uveitis flare within 2 months of vaccination, three of them happened within two weeks and were easily controlled by topical steroids without adding systemic immunosuppressive therapies. All the flares exhibited as an increase in the anterior chamber cell counts, without posterior segment inflammatory aggravation. The detailed information of the 14 patients is described in the Appendix A. All patients who suffered from the flare were surveyed afterwards and patients claimed a similar flare rate based on their experience of previous episode frequency. Naranjo criteria were also used to test the causality, and all patients had a score less than 4. All flares after vaccination were controlled without causing permanent visual acuity damage.

## 4. Discussion

Many cases of ocular events have been reported after COVID-19 vaccinations, such as facial nerve or abducens nerve palsy, acute macular neuroretinopathy, central serous retinopathy, multiple evanescent white dot syndrome, uveitis, and Graves’ Disease [8,10,22,23,24,25,26,30,31], among which the most common is uveitis. Since these inflammatory-associated cases followed soon after vaccination, it raises suspicion that COVID-19 vaccine may be related to the onset of uveitis. This makes uveitis doctors and patients worry that vaccines may cause uveitis, however, the causality is difficult to determine. According to the World Health Organization–Uppsala Monitoring Centre (WHO–UMC) causality assessment criteria, the determination of a certain causality requires that the event happened within a plausible time relationship to drug intake, the event cannot be explained by disease or other drugs, response to withdrawal pharmacologically or pathologically plausible, the event is definitive pharmacologically or phenomenologically, and rechallenge satisfactory if necessary. The more criterion that are satisfied, the more likely a causality association could be defined. Although new-onset of uveitis is not uncommon after COVID-19 vaccination, the relationship might be only casual instead of causal based on the criteria, especially considering the generalization of COVID-19 vaccination.

In addition, the relationship of vaccination with new-onset of uveitis is different from that with uveitis flare in patients with stable uveitis. The latter issue is actually a real concern for patients with chronic or recurrent uveitis. To address this question, we retrospectively investigated the flare rate occurring within 30 and 60 days of inactivated COVID-19 vaccines and compared it with the flare rate occurring within randomly selected dates in patients with uveitis. To the best of our knowledge, this is the first case-controlled study regarding the association of uveitis flare and inactivated COVID-19 vaccines. The 30-day time point was selected as most uveitis patients were followed up approximately every 4 weeks in our clinic, thus, mild flare without complaint of visual symptoms could be captured. Extending the observation period to 60 days could ensure that any delayed response could also be detected. We also analyzed the flare cases using Naranjo criteria, a scoring system to assess whether there is a causal relationship between a clinical event and a drug. The results are not supportive that the vaccine is the cause of flare, however, interestingly, most of the flares happened after the second or third dose of vaccination. Further studies, including animal models, are suggested to determine if there is any reason behind this phenomenon. Basically, most of the common etiologies of uveitis were covered in this study. Age, sex and anatomical classification distributions were also representative for the overall uveitis patients [32,33]. The possible association of uveitis with vaccination reported by cases reports could not be proved in our case-controlled study. This story is similar to SLE and multiple sclerosis patients. Cases of disease onset or exacerbation with vaccination are always reported [34,35] in SLE and multiple sclerosis [36,37,38,39], however, when case-controlled or cohort studies were conducted, the result always showed no evidence of association [40,41,42,43].

Immunosuppressive therapy is also a major concern when uveitis patients receive vaccines. In this study, we also analyzed the uveitis flare rate in the vaccination group with or without immunosuppressive therapy. We showed there is no statistically significant difference in the tendency of uveitis flare in patients with different immunosuppressive treatments. This finding is in accordance with studies regarding other autoimmune disease flare and immunosuppressive treatments [2,44], confirming the safety of inactivated vaccinations for uveitis patients under immunosuppressive treatments.

Some limitations to our study should be addressed. First, this is a retrospective study, patients with flares would be more likely to be regularly followed-up, though this would cause equal effects in both groups. Secondly, even though the sample size is much more convincible than case series, studies with a larger sample size would be more favorable to increasing the confidence level of our conclusion.

## 5. Conclusions

In conclusion, there is no evidence that inactivated COVID-19 vaccines are associated with increased uveitis flare risk in patients with uveitis, no matter if they are under systemic immunosuppressive treatments. We believe that patients with uveitis in their inflammatory quiescent period should be encouraged to receive the inactivated COVID-19 vaccine according to the recommendations given by the local health authorities.

## Figures and Tables

**Table 1 vaccines-10-01680-t001:** Demographic and the clinical features of uveitis patients.

Parameters	Vaccination Group N = 60	Control Group N = 60	*p* Value
Age (mean ± SD)	39 ± 14	38 ± 16	0.514
Sex (male to female ratio)	29:31	30:30	0.715
Disease course (years)	6.49 ± 6.57	4.34 ± 5.32	0.054
BCVA at quiescent period ^#^ (median, IQR)	0.222 (0,0.460)	0 (0,0.222)	0.894
Anatomical classification (no./%)			0.091
Anterior uveitis	27/45.0	17/28.3	
Intermediate uveitis	2/3.3	0/0	
Posterior uveitis	3/5.0	5/8.3	
Panuveitis	28/46.7	38/63.4	
Etiological classification (no./%)			0.001
Bechet’s disease	8/13.3	18/30.0	
VKH	7/11.7	14/23.3	
Fuchs uveitis syndrome	7/11.7	0/0	
PSS	4/6.6	0/0	
TINU	1/1.7	0/0	
Sarcoidosis related uveitis	1/1.7	2/3.3	
Idiopathic *	32/53.3	26/43.4	

* Uveitis that could not identify a specific etiology is classified as idiopathic. ^#^ BCVA at quiescent period refers to BCVA at the nearest visit to vaccination date, which detected no active inflammation. SD: standard deviation, BCVA: best-corrected visual acuity, VKH: Vogt–Koyanagi–Harada disease; PSS: Posner–Schlossman syndrome; TINU: tubulointerstitial nephritis and uveitis syndrome; IQR interquartile range.

## Data Availability

Not applicable.

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
