# Peer review of "There Is No Evidence That Inactivated COVID-19 Vaccines Increase Risks of Uveitis Flare"

_vaccines, 2022, doi:10.3390/vaccines10101680_

Round 1

Reviewer 1 Report

1Abstract

1.       with uveitis- all types of uveitis?

2.       Fourteen patients had uveitis flare – It is preferable to specify if it was the same uveitis or whether it was a different one for example anterior uveitis had recurrent anterior uveitis or patient had anterior uveitis after posterior uveitis

3.       Most of the published studies include vaccination related flare up till 30 days.

Article

1.       COVID-19  expand before abbreviation, preferable not to start the sentence with an abbreviation

2.       On one hand, immunosuppressive treatments may increase their vulnerability to COVID-19 infection and compromise the immunological defense when unfortunately infected; reference?? Are there any published data to show this? Or they are only postulates

3.       Reactivation of quiescent anterior uveitis after COVID-19 has been reported many years later -Sanjay S, Mutalik D, Gowda S, Mahendradas P, Kawali A, Shetty R. "Post Coronavirus Disease (COVID-19) Reactivation of a Quiescent Unilateral Anterior Uveitis". SN Compr Clin Med. 2021;3(9):1843-1847. doi: 10.1007/s42399-021-00985-2. Epub 2021 Jun 7 so theoretically a flare up may be possible after vaccination.

4.       In introduction need to mention what are the inactivated vaccines available in the world, how many cases of uveitis have been reported following their administration should be mentioned.

5.       macula edema- macular

6.       INF- α- expand please

7.       What was the duration between the doses?

8.       Did the patients stop immunosuppression or adalimumab before after the vaccination according to the rheumatology association guidelines?

9.       In supplementary data- please remove the word easily, why most cases occurred after 2nd or 3rd and not after 1st dose

10.   What was the duration for which topical steroids were administered and in what dose/frequency

11.    Are the references 24, 25 exhaustive to report all ocular adverse events after vaccination?

12.   Did you test the Naranjo criteria for your patients who had flare up?

13.   Extending the observation period to 60 days could ensure that any delayed response could be also detected.- Probably most published articles do not consider upto 60 days as related to vaccination

14.   Please include images of reactivation of anterior/posterior uveitis/ panuveitis.

Reviewer 2 Report

Authors should add in tab S1 of Supplementary data , two columns that specify the patient's symptoms and the type of flare /anterior, posterior etc.) authors should simply add a column in which they explain in tab S1 the characteristics of the flare presented by patients.
For
example if it is an anterior or posterior inflammation or the reactivation of vasculitis or cystoid macular edema etc.

Author Response

Thank you for your suggestion. We have amended sour Supplementary data according to your advice.